# A system for tracking whisker kinematics and whisker shape in three dimensions

**Rasmus S. Petersen** *⊗, **Andrea Colins Rodriguez** ⊗, **Mathew H. Evans,**
**Dario Campagner, Michaela S. E. Loft**

Division of Neuroscience and Experimental Psychology, University of Manchester, Manchester, United Kingdom

⊗ These authors contributed equally to this work.
* R.Petersen@manchester.ac.uk

## Abstract

Quantification of behaviour is essential for biology. Since the whisker system is a popular model, it is important to have methods for measuring whisker movements from behaving animals. Here, we developed a high-speed imaging system that measures whisker movements simultaneously from two vantage points. We developed a whisker tracker algorithm that automatically reconstructs 3D whisker information directly from the 'stereo' video data. The tracker is controlled via a Graphical User Interface that also allows user-friendly curation. The algorithm tracks whiskers, by fitting a 3D Bezier curve to the basal section of each target whisker. By using prior knowledge of natural whisker motion and natural whisker shape to constrain the fits and by minimising the number of fitted parameters, the algorithm is able to track multiple whiskers in parallel with low error rate. We used the output of the tracker to produce a 3D description of each tracked whisker, including its 3D orientation and 3D shape, as well as bending-related mechanical force. In conclusion, we present a non-invasive, automatic system to track whiskers in 3D from high-speed video, creating the opportunity for comprehensive 3D analysis of sensorimotor behaviour and its neural basis.

**Data Availability Statement:** Data is held in Figshare http://doi.org/10.6084/m9.figshare.9758894.v1. Code is available from https://github.com/PetersenLab/WhiskerMan.

## Author summary

The great ethologist Niko Tinbergen described a crucial challenge in biology to measure the "total movements made by the intact animal"[1]. Advances in high-speed video and machine analysis of such data have made it possible to make profound advances. Here, we target the whisker system. The whisker system is a major experimental model in neurobiology and, since the whiskers are readily imageable, the system is ideally suited to machine vision. Rats and mice explore their environment by sweeping their whiskers to and fro. It is important to measure whisker movements in 3D, since whiskers move in 3D and since the mechanical forces that act on them are 3D. However, the computational problem of automatically tracking whiskers in 3D from video has generally been regarded as prohibitively difficult. Our innovation here is to extract 3D information about whiskers using a two-camera, high-speed imaging system and to develop computational methods to reconstruct 3D whisker state from the imaging data. Our hope is that this study will facilitate

**Funding:** This work was funded by Biotechnology and Biological Sciences Research Council grants (BB/L007282/1, BB/ P021603/1) and Medical Research Council grants (MR/L01064X7/1, MR/ P005659/1) to RSP and a CONICYT Becas Chile contract 72170371 to ACR. The funders had no role in study design, data collection and analysis, decision to publish, or preparation of the manuscript.

**Competing interests:** The authors have declared that no competing interests exist.

comprehensive, 3D analysis of whisker behaviour and, more generally, contribute new insight into brain mechanisms of perception and behaviour.

## Introduction

Substantial progress towards the long-standing ambition of measuring "total movements made by the intact animal" [1] is coming from the application of machine vision methods to video recordings of behaving animals [2]. Since the whisker system is a major experimental model and since the whiskers are readily imageable [3,4], the whisker system is ideally suited to this endeavour. Tracking the whiskers of mice/rats has already deepened our understanding of active sensation and refined our capacity to relate behaviour to neural mechanisms [5–14]. Our aim here was to develop a method to track whisker movements and whisker shape in 3D in behaving mice at millisecond temporal resolution.

Both the movement of whiskers and the mechanical forces of whisker-object contact are 3D. During each whisking cycle, the whisker follicles translate with respect to the head and each whisker rotates in 3D. Although only the horizontal component of this movement is typically measured, whiskers also move vertically [15] and rotate around their longitudinal axes ('roll') [5]. The mechanical forces of whisker-object contact that are the drivers of neural activity are also 3D. When a mouse 'whisks' against an object, the whiskers bend. Again, although only the horizontal component of bending is typically measured, bending can occur in all directions [16]. In the trigeminal ganglion, all directions of deflection are represented [17–20], indicating that 3D bending information is both encoded and transmitted to the brain.

Starting with the first "cinematographic" study of whisking by Welker in 1964, there is a 50 year history of increasingly sophisticated efforts to measure whisker movement from behaving animals [4]. Most studies have measured whisker movement only in the horizontal plane, using either linear, Charge Coupled Device (CCD) arrays [21,22] or high-speed imaging [6,23–25]. However, horizontal plane imaging provides direct measurement of only one of the 3 angles that define 3D whisker orientation. Moreover, estimates of whisker bending moment obtained by imaging apparent curvature of a whisker in the horizontal plane [24,26,27] can be contaminated by roll [5]. This is significant, since bending moment is a primary driver of contact-related mechanotransduction [11,13,14]. High-speed cameras sufficient to form the basis of a 3D whisker imaging system have long been available: the main bottleneck to achieving 3D whisker tracking has been the computational complexity of the 3D reconstruction problem. A few studies have measured aspects of 3D whisker movement in vivo [5,15,16,28] and ex vivo [29], but no automatic approach has so far been developed that measures both 3D whisker orientation and 3D whisker shape from high-speed video of behaving animals. Here, we obtained 3D information noninvasively from head-fixed mice using a high-speed imaging system consisting of two cameras, positioned to minimise occlusion, and developed computational methods to reconstruct 3D whisker state. We used the system to track up to 8 whiskers in parallel, and to obtain a 3D description of each whisker, encompassing both its 3D orientation and 3D shape.

## Results

### 3D imaging of whisking behaviour

To obtain a video data set with which to develop 3D whisker tracking, we trained head-fixed mice to detect objects with their whiskers (n = 6). On each trial, a vertical pole was presented

in either an anterior location out of reach of the whiskers ('no-go trial') or a posterior location within reach ('go trial'). Mice learned to perform the task accurately (81±17%, mean ± SD over mice) and performed 135±22 trials per daily session. When the pole moved up at the onset of a trial, mice would typically commence exploratory whisking. On go trials, one or more whiskers typically contacted the pole; on no-go trials, there was no contact. In this way, we obtained a varied data set, which included episodes of whisking both with and without contact (average behavioural session ~0.5M video frames).

We recorded high-speed video of mice using a system of two high-speed cameras (Fig 1). One camera imaged whisking in the horizontal plane. The other camera imaged in an off-vertical plane, the orientation of which (25° off coronal, 10° off horizontal) was optimised to minimise occlusion of whiskers against the background of the body of the mouse (Fig 1). For brevity, we refer to this second image plane as 'vertical'.

## Reconstructing whiskers in 3D from 2D views

Using the two-camera set-up, we imaged mice (1000 frames/s) as they performed the pole detection task. This resulted in a time series of image pairs (horizontal and vertical views): we refer to each such image pair as a 'frame'. Our next aim was to develop an automated algorithm to track multiple whiskers in 3D. For two reasons, we focussed on the basal segment of the whisker shafts. First, during whisker-object contact, whiskers bend and the associated

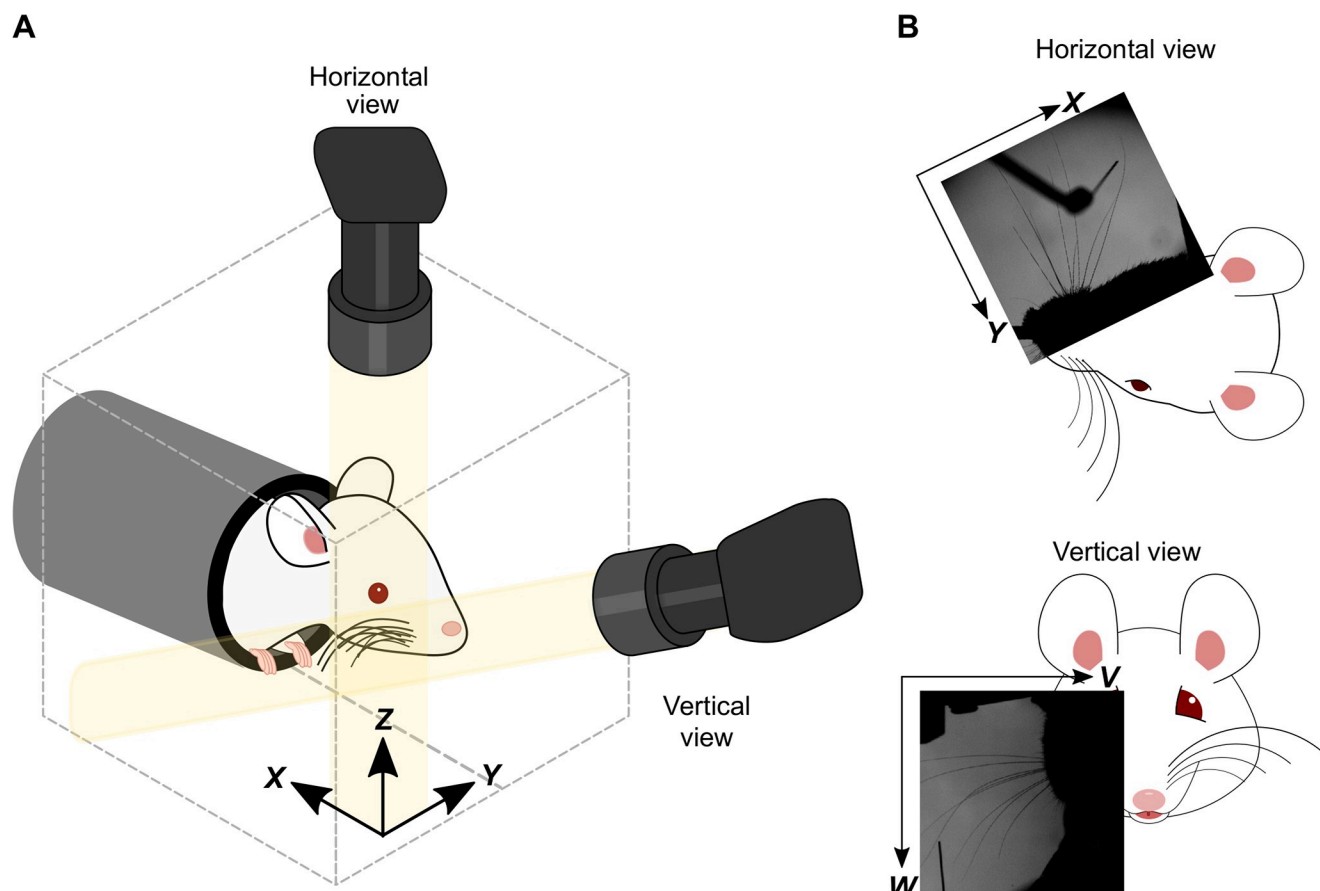

**Fig 1. Experimental set-up for 3D imaging. A**) Schematic showing the camera angles and 3D head-centred *xyz* coordinate frame. **B**) Horizontal and vertical views, with corresponding 2D coordinate frames.

mechanical forces/moments drive mechanoreceptors located in the follicle. Thus, changes in whisker shape at the base of the whisker are intimately related to neural activity in the ascending whisker pathway [11,14]. Second, tracking only the basal segment (not the whole whisker) reduces the number of parameters required to describe the shape of a whisker. To accurately describe the shape across its entire length requires at least a $5^{th}$ degree polynomial [27] which, in 3D, has 15 parameters. However, the basal segment of a whisker is well-approximated by a quadratic curve [24,29] which, in 3D, has 9 parameters.

The basic computational problem is to reconstruct the 3D coordinates of multiple whiskers from two 2D images. 3D whisker reconstruction is particularly challenging, since whiskers are similar to each other and their images can overlap. Also whiskers lack intrinsic features which can be matched across the two images. Knutsen *et al* overcame this by marking the shaft of one whisker with spots of dye [5]. The two images of each dye spot could then be matched and, after camera calibration, their 3D locations reconstructed. This is an elegant solution but there is the possibility, particularly for thin whiskers such as those of a mouse, that application of dye perturbs whisker mechanical properties. In a more recent, non-invasive approach, Huet *et al* achieved 3D tracking of a single whisker; tracking automatically in one image plane and tracking manually in a second, orthogonal plane [16]. Since there was only one whisker, coordinates describing the whisker in one plane could be uniquely matched to those in the other plane. However, manual tracking is impractical for large data sets [6,7,9,11,14]. In sum, no method currently exists that achieves automatic and non-invasive tracking of multiple whiskers in 3D. Our innovation here was to achieve this by applying constraints expressing 'prior knowledge' of natural whisking to the 3D reconstruction problem.

Our whisker tracker describes each target whisker as a 3D Bezier curve (Fig 2). This is a parametric curve segment $\mathbf{b}(s) = (x(s), y(s), z(s))$, where $0 \leq s \leq 1$ parameterises location along the curve. In our case, $s = 0$ marked the end closest to the whisker base and $s = 1$ marked the end furthest from the base. A Bezier curve is defined by 2 or more 'control points' (Fig 2), the number of which controls the complexity of the curve. We used quadratic Bezier curves, each of which has 3 control points since, as noted above, this is the lowest-degree curve that is accurate for our purposes.

## Whisker tracking algorithm

The essence of our algorithm was to track one or more target whiskers by fitting 3D Bezier curves to the image data. The core principle was, for each frame, to tune the control points of

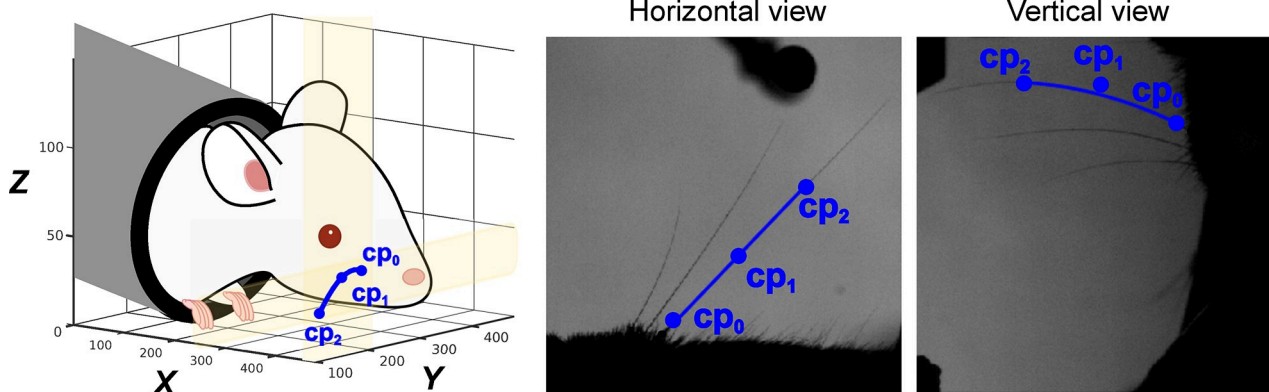

**Fig 2. Description of whiskers by quadratic 3D Bezier curves.** Left: schematic of a 3D Bezier curve representing a whisker (blue line), defined by its three control points $cp_0$, $cp_1$ and $cp_2$ (blue dots). Middle, right: projection of the 3D Bezier curve, and its control points, onto horizontal and vertical image planes.

the Bezier curves so that their projections onto the two image planes matched as closely as possible the images of the basal segments of the target whiskers. The degree of match was quantified by the following cost function (Methods, Eq 4):

$$E(f) = E_h(f) + E_v(f) + R_1(f) + R_2(f)$$

Here $E_h$ and $E_v$ (defined in Eqs 5 and 6) measured the average image intensity (the line integral) along a given Bezier curve projected into the horizontal and vertical image plane respectively; $R_1$ and $R_2$ (defined in Eqs 7 and 8) were regularising terms. Since whiskers imaged as black (low pixel intensity) and background as white (high pixel intensity), $E_h$ and $E_v$ were low when a Bezier curve $\mathbf{b}(s)$ coincided with a whisker (Fig 2), and at a local minimum with respect to local changes to the positions of the control points. In contrast, $E_h$ and $E_v$ were high if they coincided with a background region. Thus, a target whisker was tracked by minimising the cost function with respect to the control point positions of the associated Bezier curve.

It was possible to track multiple whiskers in parallel by taking advantage of prior knowledge of natural whisker motion. First, since whiskers move and deform smoothly over time, the location of a whisker in a given frame was (with sparse exceptions, see below) predictable from that in the previous frames. Thus, by seeding the control point positions of a Bezier curve representing a given whisker using corresponding positions from previous frames (Methods), it was possible to maintain accurate 'locking' between each Bezier curve and its target whisker. Second, we used prior knowledge to constrain the cost function. As detailed in Methods, a 'temporal contiguity' constraint ($R_1$) penalised discontinuous, temporal changes in Bezier curves (Methods, Eq 7) and a 'shape complexity' constraint ($R_2$) penalised unnaturally complex curve shapes (Methods, Eq 8). The full whisker tracking pipeline (Fig 3) is detailed in Methods.

## Tracking multiple whiskers in 3D

To test the algorithm, we applied it to the image data from our task. We found that we were able to track several whiskers at the same time (Fig 4; S1 Movie). Fig 4C shows a 12 ms sequence of whisker-pole contact from a mouse where 8 whiskers were intact and the others had been trimmed to the level of the fur (Fig 4A and 4B). The algorithm successfully tracked changes in both orientation and shape of the 8 whiskers. Different types of motion were tracked: some whiskers bent against the pole whilst others slipped past it (Fig 4D). The outcome of the tracker was a sequence of 3D curve segments, each representing the basal segment of a given whisker in a given frame (Fig 4E).

To assess tracking accuracy, we tracked a randomly selected set of 100 trials (50 go, 50 no-go) where a mouse was performing the task with 3 whiskers, all others trimmed to the level of the fur. This dataset comprised 350,000 frames. During 'free whisking', changes in whisker position/shape were entirely due to whisking motion, and such changes were smooth as a function of time, so that the 'temporal contiguity' and 'shape complexity' constraints of the cost function (Methods, Eqs 6 and 7) were accurate. Such errors as did occur were mainly due to (1) occlusion and (2) whisker overlap. (1) On occasion, a whisker was occluded against either the mouse's body (ear or cheek) or the experimental apparatus (pole). However, by optimising the view angles of the cameras (see above) and by minimising the image footprint of the apparatus, we minimised these effects. On the no-go trials, we detected 0.11 occlusion events/whisker per 1000 frames. Since occlusion was rare, such events were dealt with by skipping affected frames and restarting tracking afterwards. (2) On occasion, whiskers overlapped each other in either horizontal or vertical view. Because the tracker applies prior knowledge of natural whisker shape/location, our algorithm was relatively robust to such events. For

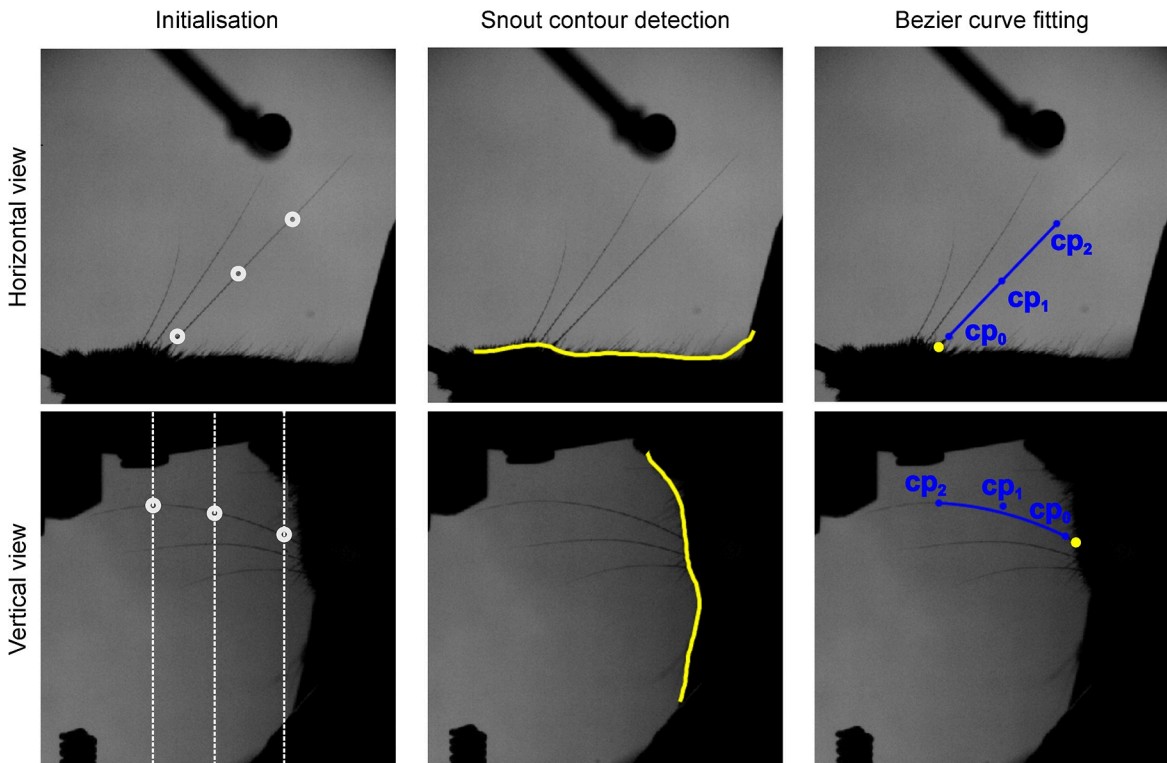

**Fig 3. Whisker tracking pipeline.** Left: Initialisation of control points for a given target whisker (see Methods for details). Initial values for control points in horizontal (top, white circles) and vertical views (bottom, white circles). White dotted lines in vertical view represent the range of $z$ values consistent with each of the $(x,y)$ points in horizontal view. Middle: Estimation of snout contour (yellow). Right: Fitting of 3D Bezier curves to image data. Projections of the 3D Bezier curve for one whisker (blue lines) and of its control points (blue dots) are shown in horizontal (top) and vertical (bottom) views. Yellow dots indicate intersections between snout contour and extrapolated Bezier curves.

example, the tracked video sequence illustrated in Fig 4 includes overlap between whiskers C1 and D2. However, errors did sometimes occur. The most common overlap error was when a small whisker overlapped a large one to such a degree that the Bezier curve tracking the small whisker locked onto the large whisker. Such events were minimised by trimming all non-target whiskers to the level of the fur. Also, since our software includes capability for manual curation, we were able to rectify these errors when they did occur, by using GUI tools to nudge the control points of the errant Bezier curve back onto its target whisker. The incidence of overlap errors increased with the number of intact whiskers. It depended also on their location: there was typically less overlap within a row of whiskers than across an arc. On no-go trials of the test data set, there were 0.01 overlap errors/whisker per 1000 frames.

Tracking was more challenging when videos included not only whisking motion but also whisker-object contact. During contact, changes in whisker position/shape from frame to frame were most often smooth and gradual but, occasionally, a whisker slipped off the pole at high speed ('slip event'), generating discontinuous whisker change between adjacent frames (Fig 4D and 4E). During such slips, tracking errors sometimes occurred, since the tracker's routine for estimating the location of a whisker based on previous frames assumes smooth motion. On go-trials, high-speed slips occurred in a small fraction of video frames (0.23 slips/whisker per 1000 frames). As above, using the software's GUI curation tools, we were able to correct errors resulting from slips.

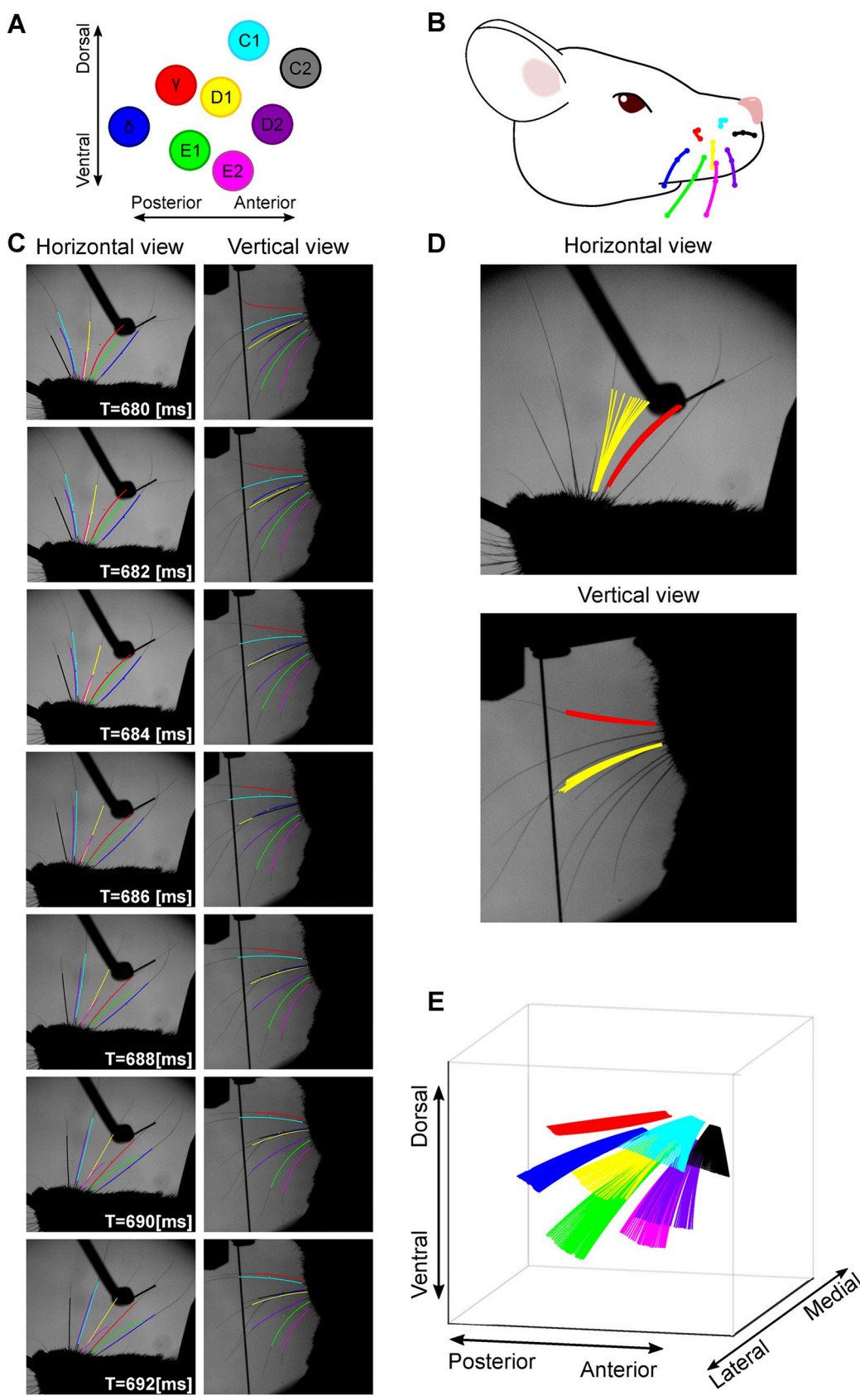

**Fig 4. Tracking multiple whiskers in 3D. A**-**B**) 8 whiskers were tracked in a 3.5 s video sequence (1000 frames/s). **C**) A sequence of 12 frames showing Bezier curves for all tracked whiskers, projected into horizontal and vertical views, taken from the example video (S1 Movie). Whiskers are colour coded as in panel **A**. **D**) Tracking solutions for 2 whiskers (colour coded as in panel **A**) across 12 frames projected onto horizontal and vertical views. **E**) 3D tracking solutions for 8 whiskers across a sequence of 30 frames, including the sequence of panel **D**.

## Measuring 3D whisker orientation and 3D whisker shape

Having tracked one or more whiskers in a set of videos, the next step was to use the tracking data to estimate 3D whisker kinematics and 3D whisker shape. To this end, we measured the 3D orientation of each whisker [5,16,29] in terms of its azimuth ($\theta$), elevation ($\varphi$) and roll ($\zeta$) (Fig 5; Methods; S2 Movie).

To illustrate the method, we first estimated 3D whisker angles during free whisking, and illustrate results for a mouse with whiskers C1-C3 intact (Fig 6). Azimuthal whisker angle was highly correlated across whiskers (whiskers C1-C3, Pearson correlation coefficients $\rho = 0.98$–0.99), as was elevation ($\rho = 0.94$–0.99) (Fig 6A). Elevation was highly anti-correlated with azimuth ($\rho = -0.89$–0.96; Fig 6B left). Roll angle correlated with azimuth/elevation but, consistent with [5], the degree of correlation was whisker-dependent ($\rho = 0.13$–0.80; Fig 6B middle,

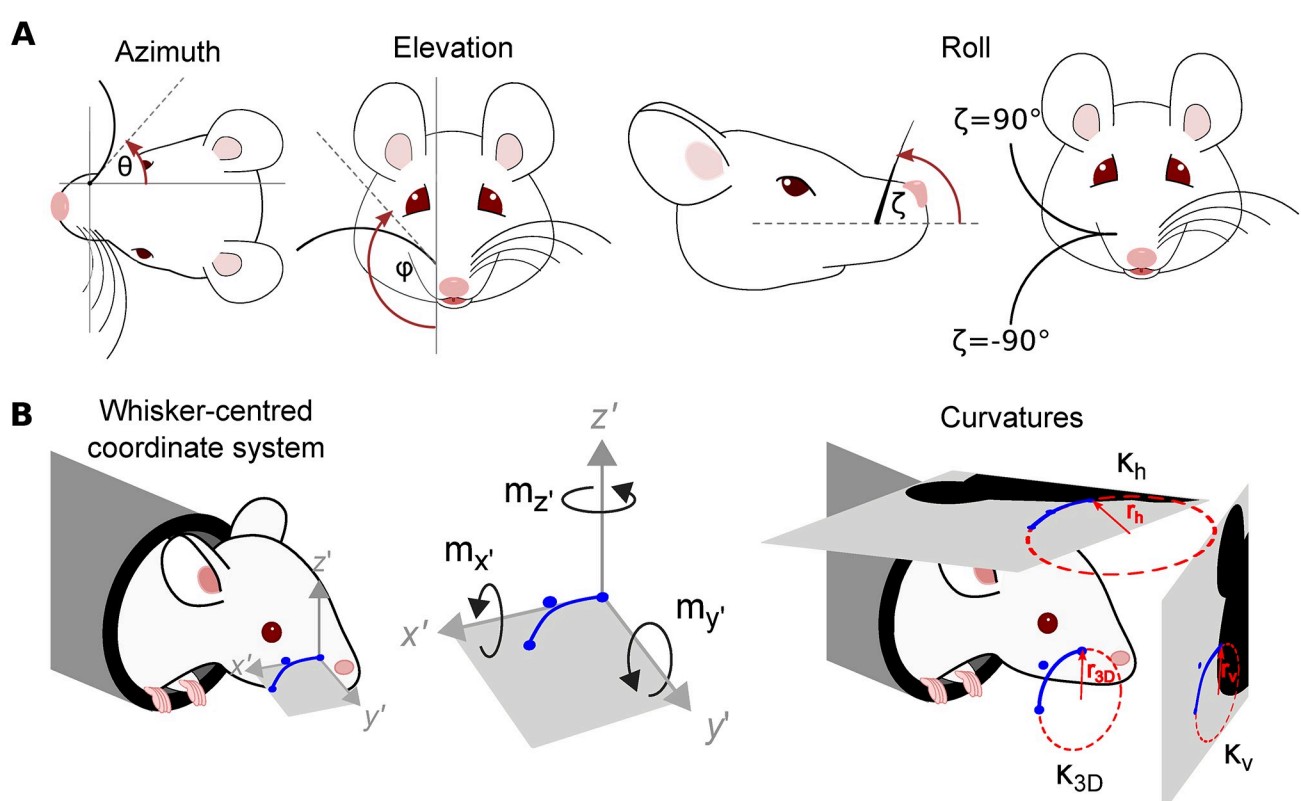

**Fig 5. Description of a whisker in terms of 3D kinematic and 3D shape parameters. A**) Azimuth ($\theta$), elevation ($\varphi$) and roll ($\zeta$) angles. These angles are defined with respect to the tangent to the Bezier curve $\mathbf{b}(s)$ describing the whisker, at $s = 0$. Azimuth describes rotation about the vertical (dorso-ventral) axis through $s = 0$; elevation describes rotation about the horizontal (anterior-posterior) axis through $s = 0$; roll describes rotation about the $x'$ axis, defined in panel **B**. **B**) *Left*. Whisker-centric coordinate frame with origin at $s = 0$ (Eqs 9–11). The $x'$ axis is tangent to $\mathbf{b}(s)$ at $s = 0$; the $y'$ axis is the direction in which $\mathbf{b}(s)$ curves; the $z'$ axis is orthogonal to the $x'$–$y'$ plane. *Middle*. Components of moment in the whisker-centric coordinate frame. *Right*. 2D and 3D whisker curvature (Eqs 13–15). $r_h$ and $r_v$ denote the radii of the circles that best fit the projection of $\mathbf{b}(s)$ into the horizontal and vertical image planes respectively (at a given point $s$); $r_{3D}$ denotes the radius of the circle that best fits $\mathbf{b}(s)$ itself.

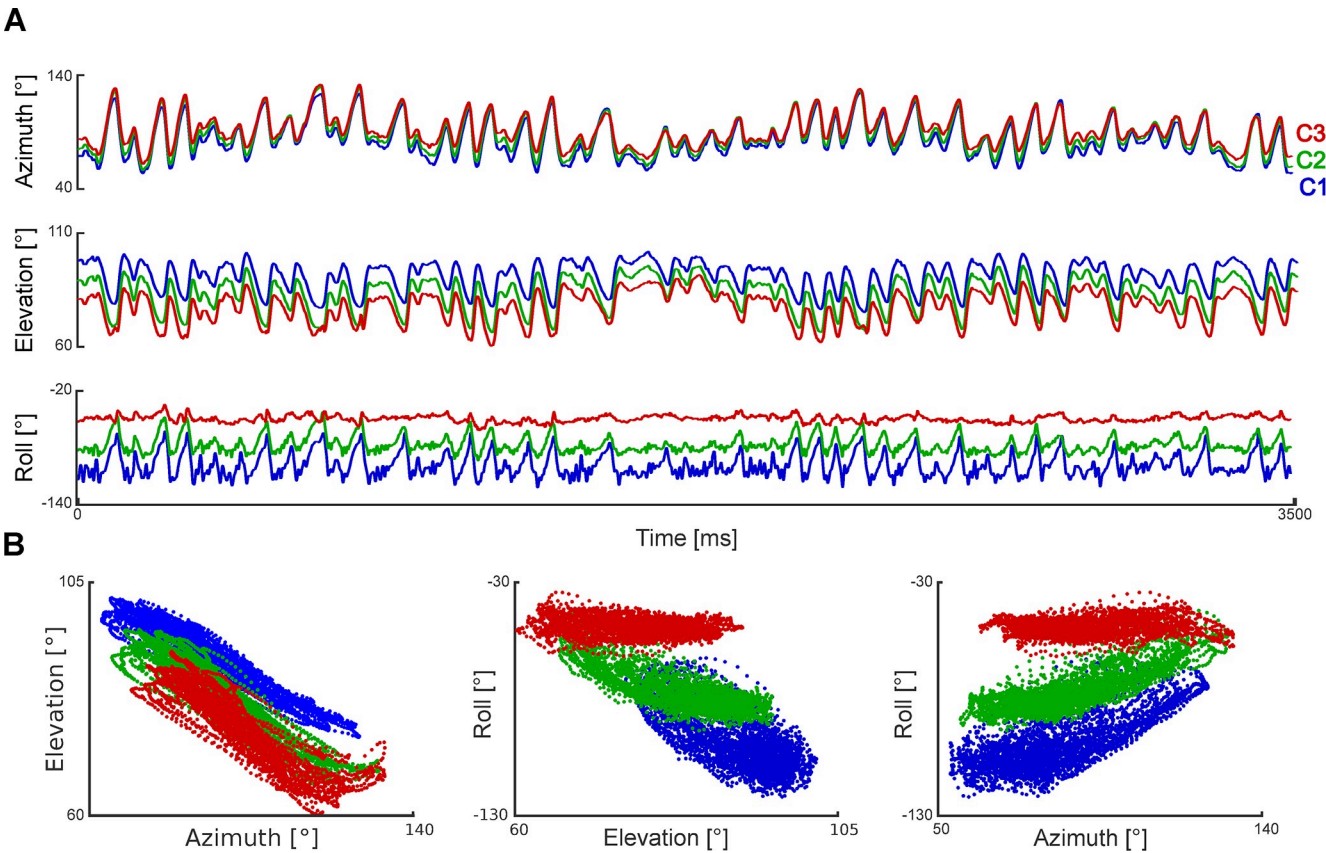

**Fig 6. 3D whisker kinematics during free whisking. A**) Changes in 3D angles for whiskers C1, C2 and C3 during a 3.5 s episode of free whisking. **B**) Relationships between angles.

right). Whisker-object contact perturbed these angle relationships and partially decorrelated the azimuth-elevation relationship ($\rho$ = -0.49–0.79).

An important feature of video-based whisker tracking is that it allows non-invasive measurement not only of kinematics but also of mechanical forces/moments acting on the whiskers [4,16,24,26]. The physical basis for this is that the shape of a whisker contains information about these mechanical factors. In particular, when a whisker quasi-statically bends against an object, there is a linear relationship between whisker curvature and bending moment. On this basis, studies combining video-based tracking with measurement of neural activity have discovered that bending moment is the best single predictor of primary afferent activity during whisker-object contact [4,11,13,14]. However, it should be noted that, in other situations (whisking in the absence of object contact and dynamic situations such as texture exploration and collisions), curvature does not have the same clean, mechanical interpretation [30–32].

Previous studies have sought to measure the bending moment related to whisker-object contact by imaging in the horizontal plane [6,7,9–11,14,24,25]. However, there are limitations of the planar imaging approach. First, it senses only the component of bending moment in the horizontal image plane and necessarily misses any out-of-plane bending. Second, since whiskers roll during the whisking cycle, the shape of a whisker, as projected in the horizontal plane can change purely due to roll even during free whisking (Fig 7). A benefit of 3D imaging is that it addresses these limitations. First, 3D imaging enables bending in any direction to be measured. Second, it permits the intrinsic shape of a whisker to be teased apart from both its

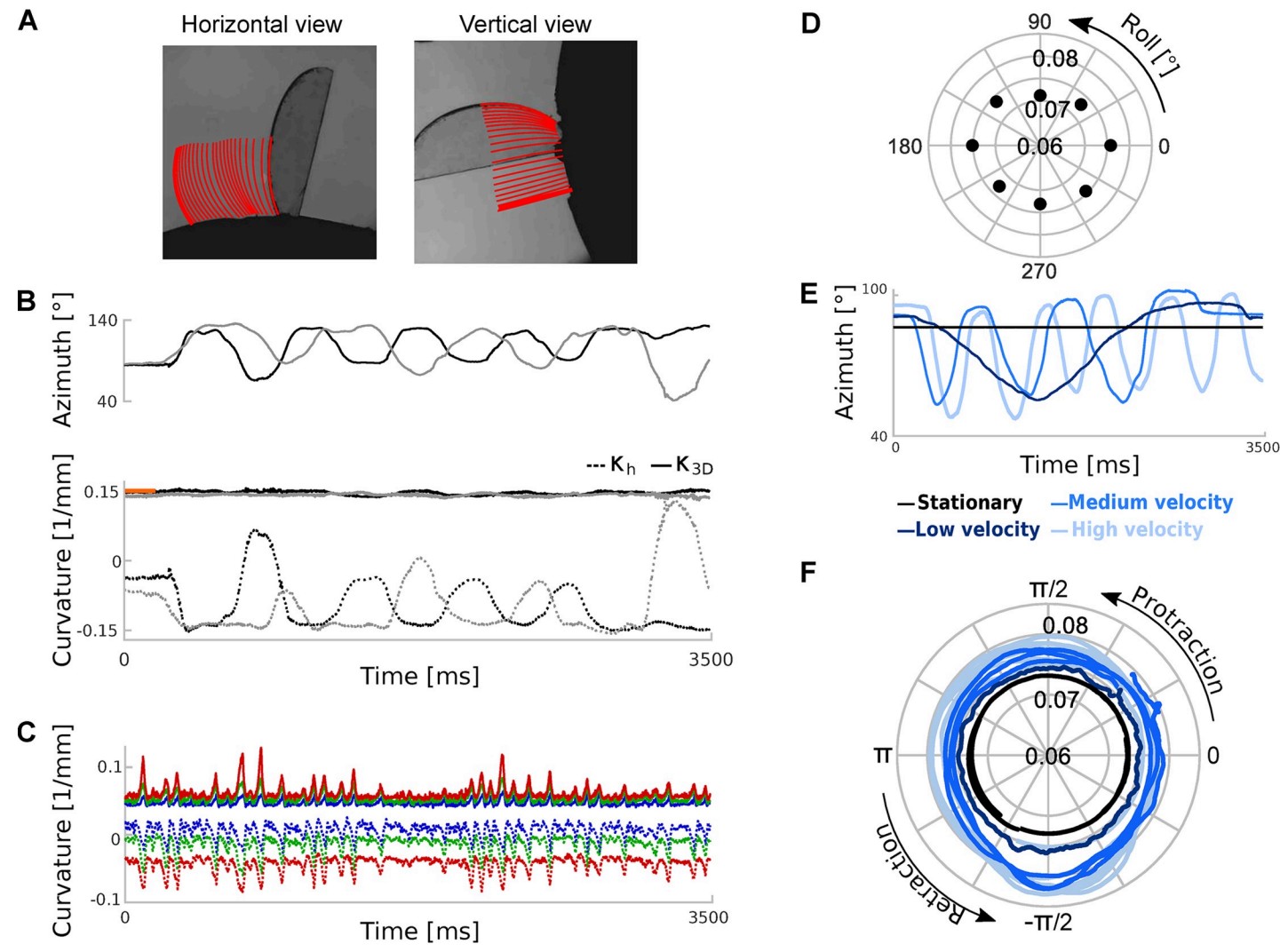

**Fig 7. Tracking and estimating 3D curvature for a rigid test object (panels A-B), whiskers of a behaving mouse (panel C) and an ex vivo whisker (panels D-F). A)** Tracking the edge of a coverslip. The coverslip was mounted, like a lollipop, on a rod; the rod was oriented in the mediolateral direction and rotated around its axis. Red lines indicate tracking results (30 frames, 10 millisecond intervals, 1000 frames/s). **B)** Top: Azimuth angle for two trials (black and grey traces). Bottom shows measured curvature: horizontal curvatures (dotted lines), $\kappa_{3D}$ (solid lines) and true curvature (orange). **C)** Horizontal and 3D curvatures during free whisking (same trial as Fig 6). Solid lines represent $\kappa_{3D}$ and dotted lines indicate horizontal curvatures for C1-3 (colours coded as in Fig 6). Fluctuations in vertical curvature were similar to those in horizontal curvature ($|\rho|>0.49$). **D)** Variation in $\kappa_{3D}$ for a stationary ex vivo whisker (C3) as a function of roll angle. **E)** Azimuth angle for ex vivo trials with simulated whisking at different speeds. **F)** $\kappa_{3D}$ as a function of whisking phase.

position and angular orientation. We measured the intrinsic shape, which we term $\kappa_{3D} : \kappa_{3D}(s)$ expresses the 3D curvature at each point $s$ along the whisker shaft and has the key property of being invariant to curve location and to curve orientation (Methods; Eq 13).

First, to verify that our system accurately measures 3D curvature we tested whether it could correctly recover the shape of a rotating, rigid object. To this end, we simulated whisking (motion with both forward-backward and rolling components) of a rigid disk, whose edge had curvature comparable to that of a typical mouse whisker (a 13 mm diameter, glass coverslip). We imaged it as described above and tracked its edge (Fig 7A). Due to the rolling motion, curvature in the horizontal and vertical planes, $\kappa_h$ and $\kappa_v$, oscillated strongly (Fig 7B). Despite this, our algorithm correctly recovered that 3D curvature $\kappa_{3D}$ was constant—fluctuations (SD)

in $\kappa_{3D}$ were 5.6% that of $\kappa_h$ and 8% that of $\kappa_v$—and matched the true curvature of the object (Fig 7B).

However, when we measured $\kappa_{3D}(s)$ of whiskers from behaving mice, we found substantial fluctuations, even during free whisking (Fig 7C): SD of $\kappa_{3D}$ was 40–91% that of $\kappa_h$ and 51–68% that of $\kappa_v$. This suggests that, in contrast to rat where the proximal segment of the whisker shaft undergoes rigid motion during whisking [5], during natural whisking, mouse whiskers do not behave as rigid objects. To determine whether this lack of rigidity occurs in the absence of whisker movement ('static') or whether it is dependent on movement ('dynamic'), we first positioned a (stationary) ex vivo mouse whisker (C3) in the apparatus at a series of roll angles, imaged it, tracked it and measured $\kappa_{3D}$. Varying roll angle changed $\kappa_{3D}$ by up to 6% (Fig 7D). Next, we 'whisked' the whisker backwards and forwards in the horizontal plane at different velocities (Fig 7E) and measured $\kappa_{3D}$ as a function of whisking phase (Fig 7F). We found that $\kappa_{3D}$ changed by up to 12% over a cycle. This change in $\kappa_{3D}$ increased linearly with whisking velocity (Fig 7F; $R^2 = 0.95$). Linear extrapolation to average whisking velocity observed during free whisking in mice performing our task (0.78 °/ms) implied a $\kappa_{3D}$ change of 66% (C3). In comparison, changes in $\kappa_{3D}$ during free whisking were 56–150% (whiskers C1-3). Overall, these data indicate that whisker motion during free whisking in mouse is non-rigid and predominantly a dynamic effect.

Estimation of $\kappa_{3D}$ allowed construction of a simple proxy to the magnitude of the bending moment, which we term $\Delta\kappa_{3D}$ (Methods). This quantity is a generalisation of the corresponding measurement from horizontal plane imaging, referred to here as $\Delta\kappa_h$. Since the mice were whisking against a vertical pole, the predominant changes in curvature were in the horizontal plane and, thus, one might expect minimal benefit from the 3D approach. Even here, however, we found $\Delta\kappa_h$ to be markedly contaminated by roll. A simple instance of this effect is shown in Fig 8A (S3 Movie). Here, the whisker initially curved downwards (roll angle -90°): whisker-pole contact from time 0–45 ms rolled the whisker in the caudal direction (roll angle -180°) with only minimal change in 3D curvature but with substantial effect on the curvature projected in the horizontal plane. Thus, $\Delta\kappa_h$ increased by 0.05 mm$^{-1}$, introducing a marked mismatch between $\Delta\kappa_h$ and $\Delta\kappa_{3D}$ (Fig 8A bottom). A more typical and complex instance of the effect of roll angle is shown in Fig 8B (S4 Movie). Here there were large fluctuations in $\Delta\kappa_h$ (e.g., at times 310–370 ms) which almost entirely reflected changes in roll angle in the absence of change to 3D curvature. On average, $\Delta\kappa_h$ explained 44% of the variation in $\Delta\kappa_{3D}$ (touch periods from 47 trials). In this way, 3D imaging permits more accurate measurement of mechanical forces acting on whiskers.

## Discussion

In order to obtain a comprehensive description of 3D whisker movements and 3D whisker-object interactions, we imaged mouse whisking behaviour using a high-speed 'stereo' imaging system. We developed software, first, to fit a 3D curve segment to each of one or more target whiskers and, second, to extract 3D kinematic and shape parameters from them. The new method allows both the 3D orientation (azimuth, elevation and roll) of a whisker and its intrinsic 3D curvature to be measured at millisecond frame rate, during both free whisking and whisker-object contact.

The vast majority of previous work on automatic whisker tracking has focussed on imaging in the horizontal plane [6,7,9,14,22–24,27,30]. However, as detailed in the Introduction, single-plane imaging necessarily captures only a fraction of the full 3D kinematic and 3D shape parameters that characterise a whisker. The advance here is a system able to extract a full 3D description of both whisker kinematics and whisker shape. The tracking algorithm is

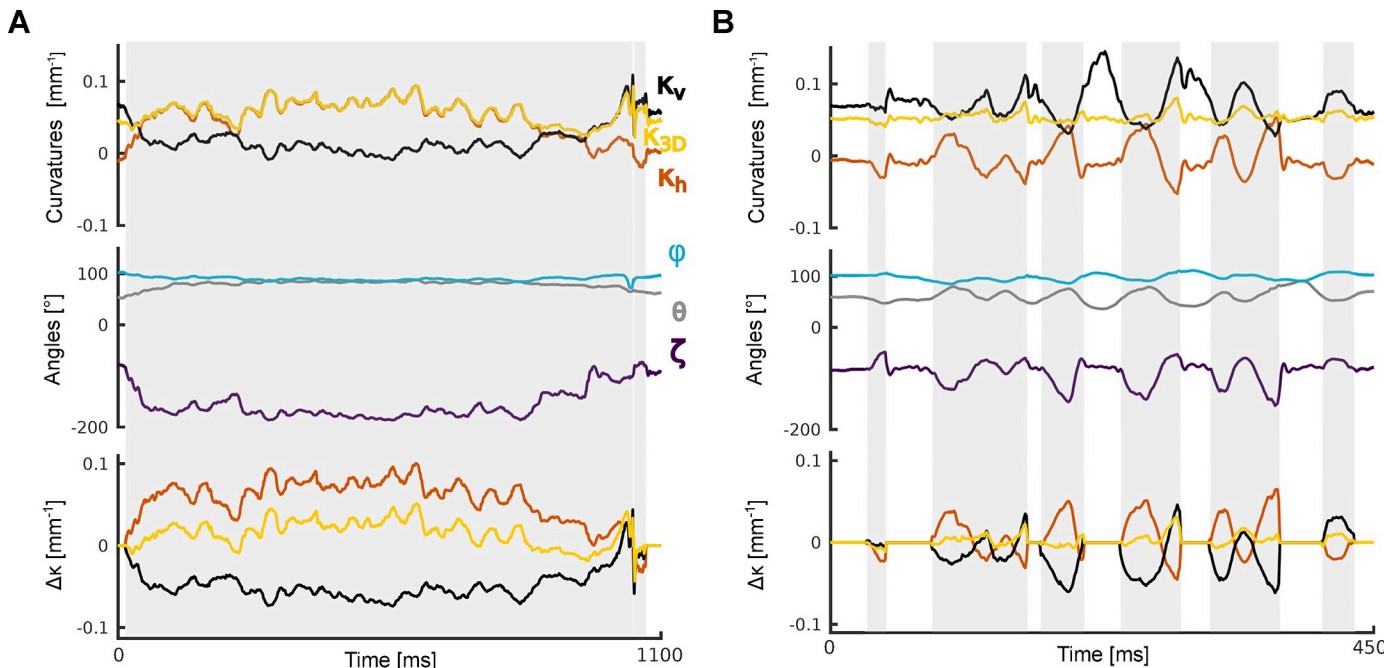

**Fig 8. Comparison of 2D and 3D curvature as mouse whisks against a pole (whisker C2): curvatures (upper panel)-, 3D kinematics (middle panel) and curvature change (bottom panel, $\Delta\kappa_{3D}$, $\Delta\kappa_h$, and $\Delta\kappa_v$). A**) Contact episode where both movement and bending of the whisker were largely restricted to the horizontal plane. In this case, $\Delta\kappa_{3D}$ and $\Delta\kappa_h$ were highly correlated. Grey shading indicates periods of whisker-pole contact. See S3 Movie. **B**) Example with same whisker as panel A for contact episode with significant vertical component of whisker motion. See S4 Movie.

automatic (after initialisation) and achieves accurate performance under our typical conditions. The system also includes GUI error correction tools, which allow accuracy to be further improved, and extend the applicability of the system to more challenging situations.

Our work builds on previous advances in 3D whisker tracking in behaving rodents. Methods based on linear CCD arrays and markers provide simple and effective means to measure some aspects of 3D whisker kinematics, but also have limitations. CCD arrays [15] image whiskers at a single point along the shaft: this method confounds whisker translation with whisker rotation, cannot measure whisker shape and cannot recover roll angle. Also, CCD study of 3D whisking has been limited to one whisker at a time. The approach of marking a whisker with spots of dye [5,28] has been used to recover all 3 orientation angles but has not been used to measure whisker shape changes during object touch. Also, although application of dye marker dots was reported not to disturb rat whisking, mouse whiskers are thinner, less stiff and hence more liable to perturbation. Our method addresses these limitations by leveraging the extra information available from video, and achieves automatic, non-invasive multi-whisker tracking, as well as the ability to recover not only three-angle kinematic information but also information about whisker shape changes during object contact. Previous 3D tracking [16] required manual tracking for the vertical view. Our algorithm runs on a standard workstation and the system only requires addition of a second camera to an existing 2D imaging set-up.

The method presented here has some limitations. First, although the error rate was low, the experimental conditions were designed to avoid both extensive object-whisker occlusion (by using a stimulus object with small foot-print) and extensive whisker-whisker overlap (by trimming non-target whiskers). Error rates are likely to be higher in the presence of experimental apparatus where there is more occlusion or if no whisker trimming is carried out. Use of additional cameras provides a potential way to reduce error rates further (and can be incorporated

through adding terms analogous to Eqs 5 and 6 to the cost function of Eq 4). Second, our tracker describes the basal segment of the whisker, not the full whisker, since our focus is primarily on elucidating the fundamental mechanical events that drive neural activity in the whisker system. To track a whisker across its entire length, a quadratic curve description is insufficient. One direction for future work is to investigate whether the tracker can be extended to fit higher degree Bezier curves without excessive loss of robustness. Third, our method has been developed for head-fixed mice. This has the advantage that it simplifies the tracking problem. However, it would also be useful to extend the approach to freely moving animals, perhaps by combining whisker tracking with head-body tracking [33–41].

In conclusion, although whisker movements are well-known to be 3D, previous, automatic methods for tracking whiskers from high-speed video were limited to the horizontal plane. Here, we obtained 3D information using a two-camera, high-speed imaging system and developed computational methods to reconstruct 3D whisker state from the video data. Our method permits measurement of both 3D whisker kinematics and whisker shape changes at millisecond frame rate from awake, behaving mice. The method can be combined with cellular resolution neural activity measurement and thus has potential to advances our understanding of sensorimotor behaviour and its neural basis.

## Methods

All parameters and variables used in this section are summarised in Table 1. All computer code was written in MATLAB (The MathWorks Inc., Natick, MA) and run on a standard workstation (Core i7, 16GB RAM).

### Ethics statement

This research has been approved by the University of Manchester Animal Welfare and Ethical Review Board and by the United Kingdom Home Office.

### Behavioural apparatus

Mice (C57; males; 6 weeks at time of implant) were implanted with a titanium head-bar as detailed in [11]. After surgery, mice were left to recover for at least 5 days before starting water restriction (1.5 ml water/day). Training began 7–10 days after the start of water restriction.

Mice were trained and imaged in a dark, sound-proofed enclosure using apparatus adapted from [11]. A head-fixed mouse was placed inside a perspex tube, from which its head emerged at one end. The stimulus object was a 0.2 mm diameter, vertical carbon fibre pole which could be translated parallel to the anterior-posterior (AP) or medio-lateral (ML) axes of the mouse by a pair of linear stepper motors and rotated in the horizontal plane to 'go' or 'no-go' locations by a rotary stepper motor. To allow vertical movement of the pole into and out of range of the whiskers, the apparatus was mounted on a pneumatic linear slide, powered by compressed air. The apparatus was controlled from MATLAB via a real-time processor. Mouse response was monitored by a lick port located anterior to the mouth. Licks were detected as described in [6]. Each lick port consisted of a metal tube connected to a water reservoir via a computer-controlled solenoid valve. Lick port position was monitored using an infrared camera and adjusted using a micromanipulator.

### Behavioural task

Head-fixed mice were trained to detect the presence of a metal pole using their whiskers, using behavioural procedures similar to [9]. On each trial, the pole was presented either within reach

**Table 1. Parameters and variables summary.**

| | | |
|---|---|---|
| **Axes** | | |
| $x$ | Anterior-posterior axis, with positive posterior | |
| $y$ | Medio-lateral axis, with positive medial | |
| $z$ | Dorsal-ventral axis, with positive dorsal | |
| $v$ | Medio-lateral axis from the vertical view, with positive medial | |
| $w$ | Dorsal-ventral axis from the vertical view, with positive ventral | |
| $x'$ | Axis of the whisker centred coordinated frame described as tangent to the whisker at $s = 0$. Positive indicates toward the tip of the whisker. | |
| $y'$ | Axis of the whisker centred coordinated frame defined by the second derivative of the Bezier curve at $s = 0$. Positive indicates $s>0$ | |
| $z'$ | Axis of the whisker centred coordinated frame defined as the cross product between $x'$ and $y'$ | |
| **Calibration** | | |
| $\mathbf{p}^H$ | Point in the horizontal plane with coefficients $(x,y)^T$ | |
| $\mathbf{H}$ | Matrix related to mapping between $\mathbf{p}^{3D}$ and $\mathbf{p}^H$. [1 0 0; 0 1 0] | |
| $\mathbf{p}^{3D}$ | Point with coefficients $(x,y,z)^T$ | |
| $\mathbf{h}$ | Vector related to the mapping between $\mathbf{p}^{3D}$ and $\mathbf{p}^H$. | |
| $\mathbf{p}^V$ | Point in the horizontal plane with coefficients $(v,w)^T$ | |
| $\mathbf{V}$ | Matrix related to mapping between $\mathbf{p}^{3D}$ and $\mathbf{p}^V$. Values of $\mathbf{V}$ are fitted during the calibration procedure | |
| $\mathbf{v}$ | Vector related to the mapping between $\mathbf{p}^{3D}$ and $\mathbf{p}^V$. Values of $\mathbf{v}$ are fitted during the calibration procedure | |
| **Bezier curves and fitting process** | | |
| $\mathbf{b}(s)$ | Bezier curve evaluated at $s$ where $0 \leq s \leq 1$ | |
| $\mathbf{cp}_i$ | Control point $i = 0,1,2$ with coordinates $(x,y,z)^T$ | |
| $j$ | whisker | |
| $f$ | frame | |
| $E_v(j)$ | Term from objective function (Eq 4) related to vertical image | |
| $E_h(f)$ | Term from objective function related to horizontal image | |
| $I_h(x,y)$ | Intensity at the point $(x,y)$ in the horizontal image | |
| $I_v(v,w)$ | Intensity at the point $(v,w)$ in the horizontal image | |
| $R_1$ | Regularising term from the objective function related to temporal contiguity | |
| $R_2$ | Regularising term from the objective function related to shape complexity | |
| $\sigma_1$ | Selectable parameter that weights the first regularising factor $R_1$ | |
| $\sigma_2$ | Selectable parameter that weights the second regularising factor $R_2$ | |
| $q$ | Vector related to Eq 8. | |
| **Extraction of kinematic parameters** | | |
| $\mathbf{i}'$ | Unit vector that point in the direction of $x'$ | |
| $\mathbf{j}'$ | Unit vector that point in the direction of $y'$ | |
| $\mathbf{k}'$ | Unit vector that point in the direction of $z'$ | |
| $\zeta$ | Rotation angle of the whisker respect to $x'$ | |
| $\theta$ | Azimuth angle defined as the angle between the $x$ axis and the projection of tangent at the base of the whisker in the horizontal plane | |
| $\varphi$ | Elevation angle defined as the angle between the $-z$ axis and the projection of tangent at the base of the whisker in the vertical image | |
| $\kappa_{3D}(s)$ | 3D Curvature evaluated at $s$ | |
| $\kappa_h(s)$ | Curvature of the projection of the Bezier curve in the horizontal plane evaluated at $s$ | |
| $\kappa_v(s)$ | Curvature of the projection of the Bezier curve in the vertical image evaluated at $s$ | |

of the whiskers ('go trial') or out of reach ('no-go trial'). At the start of each trial, the computer triggered the pole to move up (travel time ~100 ms). The pole stayed up for 1 s, before moving down. On go trials, the correct response was for the mouse to lick a lick port. Correct responses were rewarded by a drop of water (~10 μl). Incorrect responses on go trials (not licking) were punished by timeout (3–5 s). On no-go trials, the correct response was to refrain from licking and incorrect responses (licking) were punished by timeout and tone (frequency 12 kHz).

### High-speed stereo whisker imaging

Whiskers were imaged, based on the methods of [11], except that, to provide 3D information, two cameras were used. The whiskers were imaged using two high-speed cameras (Mikrotron LTR2, Unterschleissheim, Germany; 1000 frames/s, 0.4 ms exposure time) via telecentric lenses (Edmunds Optics 55–349, Barrington, NJ) as illustrated in Fig 1. Illumination for each camera was provided by a high-power infrared LED array (940 nm; Roithner LED 940-66-60, Vienna, Austria) via diffuser and condensing lens. The imaging planes of the two cameras were horizontal (spanning AP and ML axes) and vertical respectively. The field of views were typically 480 x 480 pixels, with pixel width 0.047 mm.

The two cameras were synchronised by triggering data acquisition off the computer-generated TTL pulse that initiated a trial. Typically, imaging data were acquired in an interval starting 0.5 s before pole onset, ending 1.8 s after pole onset. To provide an independent check that data files from the two cameras came from corresponding trials, an IR LED was positioned in the corner of the field of view of each camera and, starting at pole onset on each trial, flashed a binary sequence that encoded the trial number. Onset of this LED signal also served to verify camera synchrony.

### Coordinate frame and calibration

To describe the location of whiskers in 3D, we used a left-handed Cartesian coordinate frame, fixed with respect to the head of the animal (Fig 1). The axes were $x$ (AP, with positive $x$ posterior); $y$ (ML, with positive $y$ medial) and $z$ (DV, with positive $z$ dorsal). In standard anatomical convention, the $x$–$y$ (AP-ML) plane was horizontal; the $x$–$z$ (AP-DV) plane sagittal and the $y$–$z$ (ML-DV) plane coronal. The column vector $\mathbf{p}^{3D} = (x,y,z)$ denotes a point with coefficients along the $x$, $y$ and $z$ axes respectively. Throughout, we denote vectors by lower-case bold (e.g., $\mathbf{p}$), scalars by lower-case italic (e.g., $s$) and matrices by upper-case bold (e.g., $\mathbf{M}$).

Analogously to stereoscopic vision, our whisker tracker reconstructs 3D whisker location/orientation from images obtained from two viewpoints—horizontal and vertical. Pixel locations in the horizontal image were defined using the $x$ and $y$ axes of the 3D frame (Fig 1). Thus, the location of a point $\mathbf{p}^H$ in the horizontal image was described by a column vector $(x, y)$. Pixel location $\mathbf{p}^V$ in the vertical image was described by a column vector $(v,w)$, defined with respect to axes $v$ and $w$ (Fig 1). Due to the orientation of the vertical-view camera detailed above, the $v,w$ coordinate frame was rotated and translated with respect to the $x,y,z$ frame. The relation between the $x,y,z$ and $v,w$ coordinate frames was determined as follows.

With a telecentric lens, only light rays parallel to the optical axis pass through the camera aperture and contribute to the image. In contrast to standard lenses, a telecentric one provides equal magnification at a wide range of object-lens distances and forms an orthogonal projection, facilitating the reconstruction of 3D objects [42]. Since the imaging was done with telecentric lenses, the mappings from 3D to the two image planes were described as orthogonal projections. The projections of a 3D point $\mathbf{p}^{3D}$ onto a 2D point $\mathbf{p}^H$ in the horizontal image

plane and a 2D point $\mathbf{p}^V$ in the vertical plane were:

$$\mathbf{p}^H = \mathbf{H}\mathbf{p}^{3D} + \mathbf{h} \tag{1}$$

$$\mathbf{p}^V = \mathbf{V}\mathbf{p}^{3D} + \mathbf{v} \tag{2}$$

$\mathbf{H}$ and $\mathbf{V}$ were 2x3 matrices; $\mathbf{h}$ and $\mathbf{v}$, 2-element column vectors. The 3D coordinate frame and the horizontal image frame had common $x$ and $y$ axes and a common $x,y$ origin: hence, $\mathbf{H}$ = [1 0 0; 0 1 0] and $\mathbf{h}$ = [0;0].

To determine the mapping from 3D to the vertical image plane ($\mathbf{V}$ and $\mathbf{v}$), we performed the following calibration procedure. Using stepper motors, we moved an object with 2 protruding pins on a 3D path through the region of the behavioural set-up where the target whiskers were typically located, and recorded a sequence of 100 corresponding images on each camera. The $z$ location of each pin was known in each image. We then tracked the tips of the 2 pins in each horizontal and vertical image frame to obtain a time series consisting of the $(x,y,z)$ and $(v,w)$ coordinates of each pin tip. Using Eq 2, $\mathbf{V}$ and $\mathbf{v}$ were then estimated by linear regression. The variance of the regression residuals was low (<0.1% of total variance for data of Fig 6).

## Bezier curve framework for whisker tracking

Our aim was to develop 'whisker tracker' software to track the orientation and shape of one or more target whiskers. The whisker tracker described each target whisker segment as a Bezier curve, since these have convenient mathematical properties (Fig 2). A Bezier curve is a parametric curve segment $\mathbf{b}(s) = (x(s),y(s),z(s))$, where $0 \le s \le 1$ parameterises location along the curve segment: in our case, $s = 0$ marked the end closest to the whisker base and $s = 1$ marked the end furthest from the base. The shape, orientation and position of a Bezier curve are determined by its 'control points', the number of which determines the complexity of the curve. We used quadratic Bezier curves, which have 3 control points $\mathbf{cp}_i$ where $i = 0,1,2$, each with coordinates $(x,y,z)$ These control points were termed "proximal" ($\mathbf{cp}_0$), "middle" ($\mathbf{cp}_1$) and "distal" ($\mathbf{cp}_2$) according to their distance from the whisker base. $\mathbf{cp}_0$ defined the location of the basal end of the whisker segment, $\mathbf{cp}_2$ the distal end and $\mathbf{cp}_1$ the shape. In terms of these control point parameters, a quadratic Bezier curve $\mathbf{b}(s)$ was expressed as:

$$\mathbf{b}(s) = \mathbf{cp}_0(1-s)^2 + 2\mathbf{cp}_1(1-s)s + \mathbf{cp}_2 s^2 \tag{3}$$

## Whisker tracking pipeline

**Overview.** Whisker tracking was operated via a Graphical User Interface (GUI). The GUI allowed a user to load a pair of corresponding videos (horizontal view and vertical view). The first step was to calibrate, as detailed above. Next, to initialise tracking, target whiskers were specified (automatically or manually) by defining approximate locations for Bezier control points. After initialisation, each video frame was processed automatically, in turn. Tracking direction could be set to be either forwards or backwards in time. First, the contour of the snout was located in both horizontal and vertical views. Second (except in the first frame), initial estimates for the Bezier control points were calculated by linear extrapolation from their locations in the previous frames. Third, each Bezier curve was fitted to the image data by adjusting its control points to minimise the cost function defined below (Eq 4). Provided the quality of fit for a given Bezier curve met a minimum threshold, tracking of that curve proceeded automatically to the next frame.

**Manual initialisation (Fig 3 left panel).**   When tracking a video for the first time, the first step was to specify the target whiskers. In the first frame of the video, the user employed a graphical user interface (GUI) to select sets of control points, specifying one or more target whiskers. For each target whisker, the user defined approximate locations for control points specifying a curve segment corresponding to the basal segment of the whisker, by making computer-mouse clicks within the video images. For each target whisker, the user first specified $(x,y)$ coordinates for the 3 control points in the horizontal view. Since the imaging geometry was described by linear equations (Eqs 1 and 2), each such point corresponded to a line in the vertical view. In the vertical view, the user specified a point $(v,w)$ along each vertical line where it intersected the target whisker. From these $(x,y,v,w)$ data, the $z$ coordinates of the control point estimates were calculated from the calibration equation (Eq 2). Once initial values for all 3 control points of a given target whisker were specified, refined estimates were calculated by the fitting procedure described below. In order to obtain a reference value for the length of each curve, the arc length of each Bezier curve was calculated. To obtain a reference value for the distance of the proximal control point from the snout, a second order polynomial was fitted to the Bezier curve and extrapolated to find its intersection with the snout contour.

**Automatic initialisation.**   Typically, an experiment will result in many videos taken of the same mouse under identical experimental conditions. Once one video was tracked using the manual initialisation procedure described above, other videos could then be initialised automatically through a template-matching approach. To initialise tracking of a new video, the user selected a previously tracked video (via the GUI). For each target whisker from this file, a sample of Bezier curve 'templates' were extracted (typically, the solution in every fifth video frame) and goodness of fit of each sample curve to the first frame of the new video was calculated (using the cost function, Eq 4). The lowest cost template was then selected. The template was refined by optimising the fit with respect to translations along both $x$ and $z$ axes (within the range ±5 pixels).

**Snout contour detection (Fig 3 middle panel).**   First, to isolate the contour of the snout in a given video frame, fine structure such as the hairs of the fur and the whiskers were removed by median filtering (5 x 5 pixels, 0.23 x 0.23 mm) of the images followed by smoothing with a Gaussian filter (SD = 12 pixels, 0.56 mm). Next, the spatial gradient of each filtered image was calculated in a direction approximately normal to the snout contour. This gradient was small, except at the edge of the snout where it had a large peak. In the horizontal image, the snout contour was estimated as a function of the $x$ coordinate by minimising the gradient with respect to $y$. In the vertical image, the snout contour was estimated as a function of the $w$ coordinate by minimising the gradient with respect to $v$.

**Bezier curve fitting (Fig 3 right panel).**   To achieve 3D tracking, we fitted 3D Bezier curves to the horizontal and vertical view image data by varying the locations of their control points so as to minimise the following cost function. Control points for each target whisker were optimised independently:

$$E(f) = E_h(f) + E_v(f) + R_1(f) + R_2(f) \tag{4}$$

Here $E(f)$ is the cost (or mismatch) between the image data of frame $f$ and the Bezier curve $\mathbf{b}(f,s) = (x(f,s),y(f,s),z(f,s))$, defined by control points $\mathbf{cp}_0(f)$, $\mathbf{cp}_1(f)$ and $\mathbf{cp}_2(f)$. $E_h(f)$ and $E_v(f)$ quantified how well $\mathbf{b}(f,s)$ described, respectively, the horizontal and vertical image data of frame $f$. $R_1(f)$ and $R_2(f)$ were regularising terms (defined below). In the following, to keep down clutter in the notation, dependence on frame and whisker is omitted except where necessary for clarity.

$E_h$ and $E_v$ were defined as line integrals over the projection of $\mathbf{b}(s)$ in the horizontal/vertical images respectively:

$$E_h = \oint_{s=0}^{s=1} ds \, I_h(x(s), y(s)) \tag{5}$$

$$E_v = \oint_{s=0}^{s=1} ds \, I_v(v(s), w(s)) \tag{6}$$

Here: $I_h(x,y)$ was the intensity at point $(x,y)$ in the horizontal view image, calculated by linear interpolation between pixel values and $I_v(v,w)$ the analogous quantity for the vertical view image; $(x(s),y(s))$ was the projection of $\mathbf{b}(s)$ in the horizontal view image and $(v(s),w(s))$ its projection in the vertical view image (Eqs 1 and 2).

Except at occasional stick-slip events, whiskers move smoothly and, when imaged at 1000 frames/s, changes in location and shape from frame to frame were usually small, particularly for the basal segment. The regularising term $R_1$ formalised this prior knowledge of natural whisking behaviour ('temporal contiguity'):

$$R_1(f) = \frac{1}{2}\sigma_1 \sum_{i=0}^{i=2} \| \mathbf{cp}_i(f) - \hat{\mathbf{cp}}_i(f)\|^2 \tag{7}$$

Here: $\mathbf{cp}_i(f)$ was the location of control point $i$ of the whisker in frame $f$ and $\hat{\mathbf{cp}}_i(f)$ was its location estimated by linear extrapolation based on its location in the previous two frames; $\sigma_1$ was a variable gain that the user could set from the GUI.

Additional regularisation was necessary to address degeneracy that could arise when tracking near-straight whiskers. Since a line segment is fully described by the location of its two ends, a straight whisker is fully defined by its proximal and distal control points–in this case, the middle control point is ill-defined. We found, in such situations, that the middle control point tended to migrate towards the whisker base and to generate high-curvature, unnatural shapes when extrapolating the curve to the snout contour (see above). To address this, we used a second regularising term which penalised deviations of the middle control point away from the midpoint between the proximal and distal control points:

$$R_{2(f)} = \frac{1}{2}\sigma_2 \left\| \frac{(\mathbf{cp}_1(f) - \mathbf{cp}_0(f))^{\mathrm{T}} \boldsymbol{q}(f)}{\|\boldsymbol{q}(f)\|} - \frac{1}{2}\|\boldsymbol{q}(f)\| \right\|^2 \tag{8}$$

Here $\mathbf{q}(f) = \mathbf{cp}_2(f) - \mathbf{cp}_0(f)$ and $\sigma_2$ was a user-adjustable gain. $R_2$ measured deviation of the component of the middle control point $\mathbf{cp}_1(f)$ (relative to $\mathbf{cp}_0(f)$) in the direction of $\mathbf{cp}_2(f)$ (relative to $\mathbf{cp}_0(f)$) away from the midpoint of the $\mathbf{cp}_0(f) - \mathbf{cp}_2(f)$ line.

Nonlinear cost functions can be difficult to minimise due to local minima. However, in the present case, due to the smooth motion of whiskers referred to above, we expected control point solutions usually to be close to their values in the previous frame. Not only was it therefore effective to use a local search strategy, where the initial value for a given control point was set by extrapolating its values from the previous two frames $(\hat{\mathbf{cp}}_i(f))$, but this also made it possible to track multiple whiskers independently. The cost function (Eq 4) was minimised (using MATLAB function 'fminunc') with respect to components of the control points. To counteract possible drift of $\mathbf{b}(s)$ along the whisker shaft over time, or change in the arc-length of $\mathbf{b}(s)$ over time, we minimised the cost function with respect to components of $\mathbf{cp}_0$ and $\mathbf{cp}_2$ normal to $\mathbf{b}(s)$ at $s = 0$ and $s = 1$ respectively. This procedure also had the advantage of reducing the number of free parameters from 9 to 7. Furthermore, after convergence in a given frame, both the arc length of $\mathbf{b}(s)$ and the distance of $\mathbf{cp}_0$ to the snout were normalised to equal their reference values set in the first frame (see above), whilst preserving curve shape.

**Error correction.** As noted above, tracking of each target whisker proceeded automatically to the next frame, so long as the cost $E$ (Eq 4) remained less than a user-defined threshold (adjustable via the GUI). Should the threshold be exceeded, for example when a Bezier curve was 'left behind' by rapid, discontinuous motion of its target whisker during a slip event, tracking of that whisker ceased. To correct such an error, the GUI had tools allowing the user to nudge control points back onto the target whisker, and to restart automatic tracking.

## Extracting 3D kinematics of the tracked whiskers

The next step was to use the tracking data to estimate 3D whisker kinematics and 3D whisker shape. Since whiskers bend during whisker-object contact, and since this contact-induced whisker bending is a fundamental driver of neural activity (see Introduction), it was important to develop a general procedure for describing 3D whisker motion, applicable to non-rigid whisker movement We separated changes to the orientation of a quadratic curve from changes to its shape in the following manner.

Formally, we described whisker orientation by the following 'whisker-centred' Cartesian coordinate frame $x'y'z'$, with origin at $s = 0$ [16]. In contrast to the head-centred coordinate frame $xyz$, the $x'y'z'$ frame is time-dependent; rotating and translating along with its target whisker. The $x'$-axis is aligned to the longitudinal axis of the whisker (tangent to $\mathbf{b}(s)$ at $s = 0$). The $y'$-axis is orthogonal to the $x'$-axis, such that the $x'-y'$ plane is that within which $\mathbf{b}(s)$ curves. The $z'$-axis is orthogonal to both $x'$ and $y'$ axes. Let $\mathbf{i'},\mathbf{j'}$ and $\mathbf{k'}$ be unit vectors that point in the direction of the $x'$, $y'$ and $z'$ axes respectively:

$$\mathbf{i'} = \left.\frac{\frac{d\mathbf{b}}{ds}}{\left|\frac{d\mathbf{b}}{ds}\right|}\right|_{s=0} \tag{9}$$

$$\mathbf{j'} = \left.\frac{\frac{d^2\mathbf{b}}{ds^2}}{\left|\frac{d^2\mathbf{b}}{ds^2}\right|}\right|_{s=0} \tag{10}$$

$$\mathbf{k'} = \mathbf{i'} \times \mathbf{j'} \tag{11}$$

Here $\mathbf{i'} \times \mathbf{j'}$ denotes the cross product of vectors $\mathbf{i'}$ and $\mathbf{j'}$. The orientation of a whisker was then described by the 3D angle of the $x'y'z'$ coordinate frame with respect to the $xyz$ coordinate frame. We translated the frames to have a common origin and then calculated the 3D rotation matrix that rotates the $xyz$ frame to the $x'y'z'$ frame [43]. This rotation can be described as the net effect of an ordered sequence of three elemental rotations with angles $\theta$ (azimuth), $\varphi$ (elevation) and $\zeta$ (roll), and was expressed as a matrix $\boldsymbol{R}(\theta,\varphi,\zeta)$. Azimuth describes rotation in the horizontal ($x-y$) plane, about an axis parallel to the $z$ axis through the whisker base; elevation describes rotation in the vertical ($x-z$) plane, about an axis parallel to the $y$- axis; roll describes rotation around the axis of the whisker shaft (Fig 5AB). We determined the angles $\theta,\varphi,\zeta$ for a given whisker at a given time point by minimising the error function:

$$\sum (\boldsymbol{R}(\theta, \varphi, \zeta)[\mathbf{i'j'k'}] - [\mathbf{ijk}])^2 \tag{12}$$

Here $\mathbf{i}$, $\mathbf{j}$ and $\boldsymbol{k}$ are column unit vectors parallel to the $x,y$ and $z$ axes and the summation is over all matrix elements.

## Extracting 3D shape and bending moment of the tracked whiskers

Having described the orientation of a whisker, the next task was to describe its shape. By 'shape', we intend those geometric properties of a curve that are invariant to its location and

orientation. As noted above, we described whiskers by quadratic curve segments, which curve entirely within a plane (geometric torsion $\tau(s) = 0$). The intrinsic shape of a quadratic curve is fully described by a curvature function $\kappa_{3D}(s)$ [44]:

$$\kappa_{3D}(s) = \frac{\left|\frac{d\mathbf{b}(s)}{ds} \times \frac{d^2\mathbf{b}(s)}{ds^2}\right|}{\left|\frac{d^3\mathbf{b}(s)}{ds^3}\right|} \tag{13}$$

Here $|\mathbf{a}|$ denotes the magnitude (2-norm) of vector $\mathbf{a}$. $\kappa_{3D}(s)$ has units of 1/distance and is the reciprocal of the radius of the circle that best fits the curve at point $s$. We computed planar curvatures as:

$$\kappa_h(s) = \left.\frac{\frac{dx}{ds}\frac{d^2y}{ds^2} - \frac{d^2x}{ds^2}\frac{dy}{ds}}{\left(\left(\frac{dx}{ds}\right)^2 + \left(\frac{dy}{ds}\right)^2\right)^{\frac{3}{2}}}\right|_{s=0} \tag{14}$$

$$\kappa_v(s) = \left.\frac{\frac{dz}{ds}\frac{d^2y}{ds^2} - \frac{d^2z}{ds^2}\frac{dy}{ds}}{\left(\left(\frac{dz}{ds}\right)^2 + \left(\frac{dy}{ds}\right)^2\right)^{\frac{3}{2}}}\right|_{s=0} \tag{15}$$

Here $x(s),y(s),z(s)$ are the components of $\mathbf{b}(s)$ in the $x,y,z$ coordinate frame. Note, as detailed in Results, that, in contrast to $\kappa_{3D}(s)$, $\kappa_h(s)$ and $\kappa_v(s)$ are not invariant measures of geometric shape; they depend also on curve orientation.

In whisker-centric coordinates, bending corresponds to changes in shape of $\mathbf{b}(s)$ in the $x'-y'$ or $x'-z'$ planes (with, respectively, component $\mathbf{m}_{z'}$ defined in the direction of the positive $z'$ axis and $\mathbf{m}_{y'}$ defined in the directions of the positive $y'$ axis) (Fig 5;[45]). Since $\mathbf{b}(s)$ is a quadratic curve, it has zero torsion and its curvature is entirely confined to the $x'-y'$ plane: $\kappa_{3D}(s)$ is the curvature in this plane; the only non-zero component of bending moment is $\mathbf{m}_{z'}$. Applying the standard relation between bending moment about a given axis and curvature in the plane normal to that axis [26,29], it follows that $\mathbf{m}_{z'}(s)$ is proportional to:

$$\Delta\kappa_{3D}(f,s) = \kappa_{3D}(f,s) - \kappa_{3D,0}(s) \tag{16}$$

where $\kappa_{3D,0}(s)$ is the curvature when the whisker is free from contact and in its resting state. All results presented here were evaluated at $s = 0$.

## Supporting information

**S1 Movie. Tracking example of 8 whiskers.** Left: horizontal view of the whiskers and tracking superimposed. Colours are shown as in Fig 4A. Axes are shown as in Fig 1B. Middle: Vertical view of the whiskers and tracking superimposed. Right: Bezier curves in the three dimensional space. Axes are shown as in Fig 1A.
(MP4)

**S2 Movie. Whisker tracking and variables.** Top: Horizontal and vertical views with tracking superimposed. Colours are shown as in Fig 4A. Bottom: 3D kinematic and 3D shape parameters: Horizontal angle (Azimuth), vertical angle (Elevation), horizontal and vertical curvature, $\kappa_{3D}$ and roll for each tracked whisker.
(MP4)

**S3 Movie. Whisker tracking example of movement restricted to the horizontal plane (Fig 8A).** Top from left to right: Horizontal and vertical views with tracking of C2 superimposed.

Bezier curve in the whisker centred coordinate frame isolating roll angle (Fig 5A). Comparison of whisker shape over time: Dashed line represents the whisker shape at $t = 0$ ms and solid line represents whisker shape of current frame. Bezier curve was rotated using azimuth, elevation and roll angle to be captured in the two dimensional plane. Bottom from left to right: Horizontal angle (Azimuth), Vertical angle (Elevation), Roll, horizontal and 3D curvature over time. Colours are shown as in Fig 8A.
(MP4)

**S4 Movie. Whisker tracking example of movement with significant vertical components (Fig 8B).** Top from left to right: Horizontal and vertical views with tracking of C2 superimposed. Bezier curve in the whisker centred coordinate frame isolating roll angle (Fig 5A). Comparison of whisker shape over time: Dashed line represents the whisker shape at $t = 0$ ms and solid line represents whisker shape of current frame. Bezier curve was rotated using azimuth, elevation and roll angle to be captured in the two dimensional plane. Bottom from left to right: Horizontal angle (Azimuth), Vertical angle (Elevation), Roll, horizontal and 3D curvature over time. Colours are shown as in Fig 8A
(MP4)

## Acknowledgments

We thank M. Marvall for comments on the manuscript. M.E. Evans' present address: Department of Automatic Control and Systems Engineering, University of Sheffield, Sheffield, UK. D. Campagner's present address: Sainsbury Wellcome Centre for Neural Circuits and Behaviour and Gatsby Computational Neuroscience Unit, University College London, London, UK.

## Author Contributions

**Conceptualization:** Rasmus S. Petersen.

**Data curation:** Andrea Colins Rodriguez.

**Funding acquisition:** Rasmus S. Petersen.

**Investigation:** Rasmus S. Petersen, Andrea Colins Rodriguez, Michaela S. E. Loft.

**Methodology:** Rasmus S. Petersen, Andrea Colins Rodriguez, Mathew H. Evans, Michaela S. E. Loft.

**Project administration:** Rasmus S. Petersen.

**Software:** Rasmus S. Petersen, Andrea Colins Rodriguez, Mathew H. Evans.

**Supervision:** Rasmus S. Petersen.

**Validation:** Andrea Colins Rodriguez, Dario Campagner.

**Writing – original draft:** Rasmus S. Petersen, Andrea Colins Rodriguez.

**Writing – review & editing:** Rasmus S. Petersen, Andrea Colins Rodriguez, Mathew H. Evans, Dario Campagner, Michaela S. E. Loft.

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
