## [Decision Letter · Decision Letter 0]

30 Oct 2019

Dear Dr Petersen,

Thank you very much for submitting your manuscript, 'A system for tracking whisker kinematics and whisker shape in three dimensions', to PLOS Computational Biology. As with all papers submitted to the journal, yours was fully evaluated by the PLOS Computational Biology editorial team, and in this case, by independent peer reviewers. The reviewers appreciated the attention to an important topic but identified some aspects of the manuscript that should be improved.

We would therefore like to ask you to modify the manuscript according to the review recommendations before we can consider your manuscript for acceptance. Your revisions should address the specific points made by each reviewer and we encourage you to respond to particular issues Please note while forming your response, if your article is accepted, you may have the opportunity to make the peer review history publicly available. The record will include editor decision letters (with reviews) and your responses to reviewer comments. If eligible, we will contact you to opt in or out.raised.

- Supporting Information uploaded as separate files, titled 'Dataset', 'Figure', 'Table', 'Text', 'Protocol', 'Audio', or 'Video'.

We hope to receive your revised manuscript within the next 30 days. If you anticipate any delay in its return, we ask that you let us know the expected resubmission date by email at ploscompbiol@plos.org.

Sincerely,

Robyn A. Grant

Guest Editor

PLOS Computational Biology

Samuel Gershman

Deputy Editor

PLOS Computational Biology

[LINK]

Thank you for submitting your manuscript.

Your manuscript has been seen by reviewers, and I am pleased to report that, subject to satisfactory minor revision, it is considered potentially acceptable for publication.

Comments from the reviewers will help to make the paper more accessible and applicable to researchers outside of your direct field of research, and also to justify some of your statements and practices.

I look forward to receiving your next version.

Reviewer's Responses to Questions

**Comments to the Authors:**

Reviewer #1: General comments:

Seems like a very nice method, nice write up too. Would like to hear a little more about applicability, maybe, to make it as useful as possible to other experimenters.

Specific comments:

The authors use feed-forward tracking rather than full-sequence (forward-backwards) tracking. Is there any comment worth making here, trade-off of algorithmic or computational complexity vs performance?

Fig 4 - nice result. Is it cherry-picked, given the discussion in lines 170-178?

l208 "In particular, there is a linear relationship between whisker curvature and bending moment." - is this true in general? E.g. for a tapered whisker model? For large bends? Citation?

Why did the authors choose three control points? Was this because this is the minimum number required to measure a bending moment? Or some other reason? l105 "is well-approximated by" is subjective, if practical.

Fig 7 I got confused about there being only one "horizontal curvature" trace per trial in the lower panel. Worth saying "kv not shown for clarity" (that's right, is it?)?

Fig 7D is tough to read - is that a polar plot with a false radial origin? Would this be easier to read as a Cartesian plot, perhaps? The actual data, rather than the axes/annotations, is rather cramped.

Fig 7F - I can see why this is plotted polar, because it's periodic, but it's still quite tough to read, I guess because it's a bit cramped, again. Moving things around a bit so that the actual plot lines are a lot larger might help, I think. If 7D is intended to marry up with this one, and that's why it's polar, then same comment there - could be reorganised to use the available space better.

l260 "In this way, 3D imaging permits more accurate measurement of mechanical forces acting on whiskers." Certainly seems that way, good stuff.

l293 "Use of additional cameras provides a potential way to reduce error rates further." Can the authors make a statement on the applicability of the current method to 3+ cameras?

Perhaps a little more discussion on applicability? Computational load? Equipment requirements? Obstacles to running this online?

Reviewer #2: review is uploaded as an attachement

Reviewer #3: The study of Petersen et al. describes a novel combination of known computer-vision technologies, and an impressive auto-tracker, to the specific problem of tracking whisker shape in 3D. While potentially suitable for publication as a Methods paper in PLoS CB, there are a few concerns.

1. As it currently stands, the manuscript gives the impression that the 3D merging works because Bezier curves were used. In fact, as indicated in Methods, the 3D merging approach is viable because the authors performed a point-to-point calibration between the cameras using two pins mounted on a motor.

It will likely be helpful to readers to explain the basic problem more clearly: in order to 3D merge, it is necessary either to ensure that several points along the whisker length can be directly matched between the two cameras, or to perform pixel-to-pixel calibration between cameras. The first approach was used by Knutsen et al., 2008, who placed dye at known points along the whisker length. The second approach was used by Huet et al., 2015 and is used in the present work. This information would help readers better compare technical approaches between studies, and would highlight the novelty of the present work, which lies in automating the approach for multiple whiskers and in making the code publicly available.

2. Because this is a Methods paper, it would be helpful for the introduction to be more clear that the approach is currently limited to the head fixed animal, to a single row of whiskers, and for very specific locations of the cameras.

3. As currently written the manuscript risks misleading the field about the relationship between whisker curvature and the mechanical signals at the whisker base. Specifically, the manuscript states: “… there is a linear relationship between whisker curvature and bending moment,” and that several studies have computed bending moment based on curvature.

But the linear relationship between curvature and bending moment holds only in quasistatic conditions. During non-contact (“free-air”) whisking and during collisions, the quasistatic approximation does not hold. To compute mechanical signals at the whisker base under these conditions requires a dynamic model of the whisker. Notably, computing whisker dynamics requires (among many other things) knowledge of the entire whisker shape all the way to the tip, not just the proximal portion.

The manuscript should add a paragraph explicitly warning the reader that even though the curvature can be computed during non-contact whisking, during texture exploration, and immediately after collision with an object, this curvature is not a good proxy for any particular mechanical signal at the whisker base. The curvature is a good proxy for bending moment only during pure bending, as the whisker deflects against an object, after vibrations from the initial collision have damped out.

Note that curvature might still correlate with a variety of neural signals, but it is not uniquely related to the mechanical signals at the whisker base (except during pure bending).

Minor:

The sentence: “Moreover, estimates of whisker-object bending force (‘bending moment’) obtained by imaging apparent curvature of a whisker…” makes it sound as though force and bending moment are synonymous. They are very different. Also the idea of “bending force” is not clear.

The sentence: “This is significant, since bending moment is the primary driver of contact-related mechanotransduction (23–25).” should be changed to “one of the primary drivers” as the axial (longitudinal) force may also play a significant role (Stuttgen and Schwartz). In addition, the transverse forces and twisting moment could also be important – no study has yet shown otherwise.

The sentence: “First, during whisker-object contact, whiskers bend and the associated mechanical force/moment drives mechanoreceptors” should be changed to forces and moments to indicate that they are distinct quantities and both plural.

**Have all data underlying the figures and results presented in the manuscript been provided?**

Reviewer #1: Yes

Reviewer #2: None

Reviewer #3: No: The code is available on github, but it is not clear where to find the data underlying figures and results on figshare.

PLOS authors have the option to publish the peer review history of their article (what does this mean?). If published, this will include your full peer review and any attached files.

Reviewer #1: Yes: Ben Mitchinson

Reviewer #2: No

Reviewer #3: No

---

## [Editor Report · Decision Letter 1]

5 Dec 2019

Dear Dr Petersen,

We are pleased to inform you that your manuscript 'A system for tracking whisker kinematics and whisker shape in three dimensions' has been provisionally accepted for publication in PLOS Computational Biology.

In the meantime, please log into Editorial Manager at https://www.editorialmanager.com/pcompbiol/, click the "Update My Information" link at the top of the page, and update your user information to ensure an efficient production and billing process.

One of the goals of PLOS is to make science accessible to educators and the public. PLOS staff issue occasional press releases and make early versions of PLOS Computational Biology articles available to science writers and journalists. PLOS staff also collaborate with Communication and Public Information Offices and would be happy to work with the relevant people at your institution or funding agency. If your institution or funding agency is interested in promoting your findings, please ask them to coordinate their releases with PLOS (contact ploscompbiol@plos.org).

Thank you again for supporting Open Access publishing. We look forward to publishing your paper in PLOS Computational Biology.

Sincerely,

Robyn A. Grant

Guest Editor

PLOS Computational Biology

Samuel Gershman

Deputy Editor

PLOS Computational Biology

---

## [Editor Report · Acceptance letter]

15 Jan 2020

PCOMPBIOL-D-19-01570R1 

A system for tracking whisker kinematics and whisker shape in three dimensions

Dear Dr Petersen,

I am pleased to inform you that your manuscript has been formally accepted for publication in PLOS Computational Biology. Your manuscript is now with our production department and you will be notified of the publication date in due course.

With kind regards,

Matt Lyles
